



# Soil organic carbon distribution for 0-3 m soils at 1 km² scale of the frozen ground in the Third Pole Regions

Dong Wang[1,2], Tonghua Wu[1,3*], Xiaodong Wu[1], Xianhua Wei[4], Cuicui Mu[5], Ren Li[1], Guojie Hu[1], Defu Zou[1], Xiaofan Zhu[1], Jie Chen[1], Junmin Hao[6], Jie Ni[1,2], Xiangfei Li[1,2], Wensi Ma[1,2], Amin Wen[1,2], Chenpeng Shang[1,2], Yune La[1,2], Xin Ma[1,2]

[1] Cryosphere Research Station on the Qinghai-Tibetan Plateau, State Key Laboratory of Cryospheric Science, Northwest Institute of Eco-Environment and Resource, Chinese Academy of Sciences, Lanzhou, Gansu 730000, China

[2] University of Chinese Academy Sciences, Beijing, 100049, China.

[3] Southern Marine Science and Engineering Guangdong Laboratory, Guangzhou 511458, China.

[4] College of geography and environmental science,Northwest Normal University,Lanzhou 730070,China.

[5] Key Laboratory of Western China's Environmental Systems (Ministry of Education), College of Earth and Environmental Sciences, Lanzhou University, Lanzhou, 730000, China.

[6] School of civil engineering, Lanzhou University of Technology, Lanzhou, 730050, China.

*Correspondence: Tonghua Wu (thuawu@lzb.ac.cn)



**Abstract:** Soil organic carbon (SOC) is very important in the vulnerable ecological environment of the Third Pole; however, data regarding the spatial distribution of SOC are still scarce and uncertain. Based on multiple environmental variables and soil profile data from 458 pits (depth of 0–1 m) and 114 cores (depth of 0–3 m), this study uses a machine-learning approach to evaluate the SOC storage and spatial distribution at a depth interval of 0–3 m in the frozen ground area of the Third Pole region. Our results showed that SOC stocks (SOCS) exhibited a decreasing spatial pattern from the southeast towards the northwest. The estimated SOC storage in the upper 3 m of the soil profile was 46.18 Pg for an area of $3.27 \times 10^6$ km$^2$, which included 21.69 Pg and 24.49 Pg for areas of permafrost and seasonally frozen ground, respectively. The mean SOCS under different vegetation types showed a decreasing pattern as follows: forest > shrub > cropland > grassland > desert. Among all soil orders, histosols and gleisoil had the largest SOCSs, while gypsisols and salt flats had the smallest SOCS. Our results provide information on the storage and patterns of SOCS at a 1 km$^2$ scale for areas of frozen ground in the Third Pole region, thus providing a scientific basis for future studies pertaining to Earth system models. The dataset is open-access and available at https://doi.org/10.5281/zenodo.4293454 (Wang et al., 2020).

## 1 Introduction

Soil is an important part of the global terrestrial ecosystem and represents the largest terrestrial organic carbon pool with the longest turnover time (Amundson, 2001). This is especially true in areas of frozen ground, including permafrost and seasonally frozen ground. In cold environments, soil accumulates substantial organic carbon due to slow decomposition rates and repeated freeze–thaw cycles (Fan et al., 2012; Li et al., 2020). It has been reported that more than half of the world's soil organic carbon (SOC) is stored in permafrost regions (Hugelius et al., 2014; Ping et al., 2015). Even slight changes in the decomposition of the SOC pool in permafrost regions might lead to significant changes in the atmospheric $CO_2$ concentration, which plays an important role in regulating and stabilizing the carbon balance of global ecosystems (Schuur et al., 2015). Therefore, it is of great significance to accurately estimate the storage and



spatial distribution of SOC in regions of frozen ground in order to study the carbon
cycle of this ecosystem as well as global change.
As the "roof of the world", the Third Pole is the area of frozen ground at the highest
average altitude in the middle and low latitudes of the Northern Hemisphere, where
permafrost and seasonally frozen ground cover areas of $\sim 1.72 \times 10^6$ km$^2$ and $\sim 1.55 \times$
$10^6$ km$^2$, respectively (Obu et al., 2019). The Third Pole is also one of the most sensitive
areas with respect to global climate change, and has a warming rate that is
approximately twice the global average (Stocker et al., 2013). In the past few decades,
permafrost in the Third Pole region has experienced obvious degradation, which is
characterized by an increasing ground temperature, a deepening of the active layer, a
shrinking permafrost area, an expanding area of seasonally frozen ground, and the
development of thermokarst (Mu et al., 2020; Ran et al., 2017; Turetsky et al., 2019;
Wu et al., 2012). Permafrost degradation will not only cause serious geological disasters
and affect engineering construction in cold areas, but will also accelerate the
decomposition of the huge SOC pool stored in permafrost. Moreover, it will emit a large
amount of greenhouse gases into the atmosphere, thus increasing the rate of climate
change in the future (Schuur et al., 2015). Therefore, accurate estimates of the SOC
storage and spatial distribution in the area of frozen ground in the Third Pole region
have become important for Earth system modeling. Such estimates are widely used to
study the carbon cycle of this ecosystem and global change (Koven et al., 2011;
Lombardozzi et al., 2016; McGuire et al., 2018).
Early studies were mostly based on data from China's national soil survey, and
were combined with regional vegetation/soil maps to estimate the SOC pool for a
certain vegetation type or relatively small area (Wang et al., 2002; Zeng et al., 2004).
Up until 2008, the Chinese part of the Qinghai-Tibet Plateau (QTP) was taken as an
independent geographical unit to estimate the SOC pool in the upper 100 cm of the soil
profile (Tian et al., 2008; Wu et al., 2008). However, these studies did not distinguish
between regions of permafrost and seasonally frozen ground. Mu et al. (2015) used data
from 11 deep sediment cores and previously published data to estimate the SOC storage
of permafrost regions on the QTP, and found this to be 27.9 Pg in the upper 2 m of the



soil profile and 132.3 Pg below a depth of 2 m. Zhao et al. (2018) used the data of 200
soil profile measurements from permafrost zones on the QTP, and reported a SOC
storage of 17.07 Pg for the upper 2 m of the soil profile. Subsequently, Jiang et al. (2019)
used the second Chinese soil census data and estimated that the total SOC pool for a
depth interval of 0–3 m on the QTP was approximately 73.61 Pg. Although the
aforementioned studies improved our understanding of SOC storage in the Third Pole
region, their results were quite different due to differences in the SOC data sources,
number of sampling sites, and research aims. Furthermore, the large-scale maps of
vegetation and soil types used in these studies were associated with large uncertainties
because they were created years ago and have a low spatial resolution, thus leading to
potentially large errors in the estimated total SOC pools.

Recently, considerable progress has been made in digital soil mapping methods.

Spatial interpolation, linear regression, and machine learning have been widely used to
simulate the spatial distribution of SOC in the permafrost region of the QTP (Ding et
al., 2016; Ding et al., 2019; Wang et al., 2020; Yang et al., 2008). These studies have
provided new spatial data and improved the prediction accuracy of SOC compared with
earlier studies. However, few studies to date have systematically assessed SOC pools
across areas of seasonally frozen ground in the Third Pole region, which limits many
investigations requiring SOC data for these areas. The average elevation of the
seasonally frozen ground in the Third Pole region exceeds 3800 m, and there is a colder
environment, longer freezing time, and slower decomposition rate of organic matter in
comparison to other regions at the same latitude (Chen and Li, 2008). In addition, the
total SOC storage cannot be neglected and requires further study.
To evaluate the size and high-resolution spatial patterns of SOC stocks in the Third
Pole region, we carried out a large-scale field-sampling plan that covered representative
permafrost zones over the region's bioclimatic gradient, including a large unpopulated
area with harsh natural conditions. A total of 200 soil pits were excavated, most of
which were deeper than 2 m (Zhao et al., 2018). In addition, we collected field-
measured SOCS data for the Third Pole region from relevant literature published
between 2000 and 2016 (Ding et al., 2016; Song et al., 2016; Xu et al., 2019; Yang et



al., 2008). By combining high-resolution remotely sensed data and interpolated meteorological datasets, we simulated the spatial distribution of SOCS in the Third Pole region by three machine learning methods and calculated the SOC storage of specific soil intervals (0–30 cm, 0–50 cm, 0–100 cm, 0–200 cm, and 0–300 cm). The results provide basic data for Earth system modeling, and reference methods for studying the spatial distribution of soil elements under complex terrain.

## 2 Materials and Methods

### 2.1 Study area

The Third Pole is the highest plateau in the world, and is located on the QTP and its surrounding mountains, which include Pamir and Hindu Kush mountain ranges in the west, the Hengduan Mountains in the east, the Kunlun and Qilian mountains in the north, and the Himalayas in the south (Yao et al., 2012). In addition, the Third Pole is the largest high-altitude permafrost zone in the Northern Hemisphere, with a total permafrost area of approximately $1.72 \times 10^6$ km$^2$, thus representing ~8% of permafrost regions in the Northern Hemisphere (Obu et al., 2019). The average active layer thickness is 2.3 m (Qin et al., 2017). The area of seasonally frozen ground covers an area of approximately $1.55 \times 10^6$ km$^2$, which is mainly located in the eastern and southern parts of the Third Pole as well as at lower elevations of basins.

Affected by high altitude, most areas of the Third Pole are dominated by a mountain plateau climate with strong solar radiation. The mean annual precipitation (MAP) ranges from 50 mm to 2000 mm and falls mainly during the growing season from May to September (Ji et al., 2018). The mean annual temperature (MAT) is < 5 °C, which gradually decreases with elevation, and has an obvious vertical climate zone (Qin et al., 2005). The Third Pole is mainly covered by five types of vegetation: forests, shrubs, grasslands, croplands, and deserts (Hao et al., 2017).

### 2.2 Data Processing

#### 2.2.1 Soil organic carbon data

The SOC data used in this study included document data and field-measured data (Table 1). 1) Document data: data pertaining to a soil depth interval of 0–30 cm (n = 135) was retrieved from Yang et al. (2010) for the SOC database. Data pertaining to a



depth interval of 0–100 cm (n = 93) was obtained from Xu et al. (2019). Data pertaining
to a depth interval of 0–100 cm (n = 30) retrieved from Song et al. (2016). Moreover,
additional data for 0–3 m and 0–2 m depth intervals (n = 113) were retrieved from Ding
et al. (2016).
2) Field measured data: a total of 200 soil pits were excavated between 2009 and
2013; 72 soil pits were excavated manually in 2009, and 128 soil pits were excavated
with hydraulic excavators in 2010 and 2011 (Zhao et al., 2018). For each soil profile,
we collected soil samples at depth intervals of 0–10 cm, 10–20 cm, 20–30 cm, 30–50
cm, 50–100, and 100–200 cm. The bulk density was measured using a bulk soil sampler
(5 cm diameter and 5-cm-high stainless-steel cutting ring). The SOC content was
determined using the Walkley-black method after soil samples were pretreated by air
drying, grinding, and screening. The analyses were carried out in triplicate using
subsamples, and the mean of three values was used as the SOC content. The SOCS was
calculated using Eq. (1):
$$SOCS = \sum_{i=1}^{n} T_i \times BD_i \times SOC_i \times \frac{(1 - C_i)}{10} \tag{1}$$

where $T_i$, $BD_i$, $SOC_i$, and $C_i$ are soil thickness (cm), dried bulk density (g·cm$^{-3}$), SOC
content (%) and > 2mm rock fragment content (%) at layer $i$.
**2.2.2 Environmental data**
The environmental covariates used in this study included a digital elevation model
(DEM), remotely sensed data, and spatial interpolation data (Table S1).
A DEM at a spatial resolution of 1 km × 1 km was downloaded from the
International Scientific Data Service Platform (http://datamirror.csdb.cn). Using the
DEM data and SAGA GIS software, we calculated 14 terrain attributes: elevation (H),
slope (S), aspect (A), plan curvature (PlanC), profile curvature (ProC), topographic
wetness index (TWI), total catchment area (TCA), relative slope position (RSP), slope
length and steepness factor (LS), convergence index (CI), channel network base level
(CNB), channel network distance (CND), valley depth (VD), and closed depressions
(CD).
Mean annual air temperature (MAT) and mean annual precipitation (MAP) data



were downloaded from WorldClim version 2.1 (https://www.worldclim.org). These
datasets were generated by organizing, calculating, and spatially interpolating observed
data from global meteorological stations for the period 1970–2000.
Normalized difference vegetation index (NDVI) data were obtained from the
United States Geological Survey (USGS) (http://modis.gsfc.nasa.gov/). The datasets
underwent atmospheric, radiometric, and geometric correction, with a spatial resolution
of 1 km × 1 km for every 1-month interval over the period 2000–2015. The NDVI
product was calculated using the maximum value composite (MVC) method, which can
minimize the effects of aerosols and clouds (Stow et al., 2004).
The net primary productivity (NPP) and leaf area index (LAI) data were obtained
from the Global Land Surface Satellite (GLASS, V3.1), which is estimated from the
MODIS reflectance data using the general regression neural network (GRNN) method
(Liang et al., 2013). Data were at a 1 km resolution for 8 day periods between 2000 and
2015, and were downloaded from the National Earth System Science Data Center of
the National Science & Technology Infrastructure of China (http://www.geodata.cn).
The soil texture data, including sand, silt, and clay contents, were obtained from
the "SoilGrids250m database" (http://www.isric.org). The original 250 m spatial
resolution data were resampled to a 1 km resolution based on nearest neighbor
interpolation using ArcGIS 10.2 software (ESRI, Redlands, CA, USA). A digitized soil
taxonomy map was provided by the Harmonized World Soil Database version 1.2
(http://www.fao.org/), which combines existing national soil information worldwide (1
km resolution).
The land cover data used in this study were collected from the Land Cover Type
Climate Modeling Grid (CMG) product (MCD12C1) from 2010
(https://lpdaac.usgs.gov). The classification schemes in this study were based on the
global vegetation classification scheme of the International Geosphere Biosphere
Programme (IGBP). We reclassified the land cover types into five major categories:
forest, shrub, grassland, cropland, and desert.
**2.3 Model predictions**
In this study, three machine learning methods (random forest (RF), gradient



boosted regression tree (GBRT), and support vector machine (SVM)) were constructed
and validated using the SOCS in the upper 30 cm of soil profiles along with associated
variables.
With respect to the machine learning methods used, RF is used for classification,
regression, and other tasks. It is operated by constructing a large number of decision
trees during training, and outputs the class as the classification or regression patterns of
single trees (Tin Kam, 1998). The GBRT method is an iterative fitting algorithm
composed of multiple regression trees, and combines regression trees with a boosting
technique to improve predictive accuracy (Elith et al., 2008). The SVM regression
method uses kernel functions to construct an optimal hyperplane, which has a minimal
total deviation (Drake and Guisan, 2006). Combined with the remotely sensed data and
spatial interpolation data, RF, GBRT, and SVM regression were conducted to predict
the SOCS in the Third Pole region. The 'randomForest', 'gbm', and 'e1071' packages
in R were used to perform RF, GBRT, and SVM analyses.
The 15 input variables (H, S, TWI, TCA, RSP, CNB, CND, VD, NDVI, NPP, LAI,
MAP, MAT, sand, and silt) for the three regression models were selected because they
can reflect the effects of topography, climate, vegetation, and soil properties on regional
SOCSs. Moreover, these variables were significantly associated with the SOCS at a
depth interval of 0–30 cm ($P < 0.01$, Table S2), whereas other environmental factors
were eliminated due to their low correlation coefficients.
To generate the spatial distributions of SOCS in deep layers (below a depth of 100
cm), we established nonlinear extrapolation models (Fig. 3.a–b; Eqs. (2)–(4)) between
the SOCS in the upper 100 cm interval and the SOCS in the upper 200 cm interval using
the data from the 200 soil pits in grassland (n = 151) and desert ecosystems (n = 49,
Fig. A1). A third extrapolation model between the SOCS in the upper 200 cm interval
and the SOCS in the upper 300 cm interval in grassland ecosystems was established
using the data from 114 sites reported by Ding et al. (2016) (Fig 3.c; Eq. (4)).
$$\ln SOCS_{0-200cm} = 0.9708 \times \ln SOCS_{0-100cm} + 0.3128 \qquad (2)$$

$$\ln SOCS_{0-200cm} = 0.8690 \times \ln SOCS_{0-100cm} + 0.7649 \qquad (3)$$



$$\ln SOCS_{0-300cm} = 0.9521 \times \ln SOCS_{0-200cm} + 0.3296 \qquad (4)$$
where $\ln SOCS_{0-100cm}$, $\ln SOCS_{0-200cm}$ and $\ln SOCS_{0-300cm}$ are the natural logarithms of the
soil organic carbon stocks (kg·m$^{-2}$) at the depth intervals of 0–100 cm, 0–200 cm, and
0–300 cm, respectively.
It is impossible to build extrapolation models directly to estimate deep SOC storage
in forest, shrub, and cropland ecosystems, which lack deep soil pits below 100 cm.
Therefore, according to the vertical distribution of the SOCS associated with different
land cover types worldwide from Jobbagy and Jackson (2000), the extrapolation models
shown in Eqs. (5)–(6) were established indirectly to estimate deep SOC storage (below
a depth of 100 cm) in areas of these land cover types (Fig. S1). Correspondingly, Eq.
(7) was established to estimate the deep SOC storage (below a depth of 200 cm) in
desert ecosystems due to a lack of deep soil pits below 200 cm.
$$SOCS_{0-200cm} = \left(1 + \beta_{100-200cm}\right) \times SOCS_{0-100cm} \qquad (5)$$
$$SOCS_{0-300cm} = \left(1 + \beta_{100-200cm} + \beta_{200-300cm}\right) \times SOCS_{0-100cm} \qquad (6)$$
$$SOCS_{0-300cm} = SOCS_{0-200cm} + \beta_{200-300cm} \times SOCS_{0-100cm} \qquad (7)$$
where $\beta_{100-200cm}$ and $\beta_{200-300cm}$ are proportion of $SOCS_{100-200cm}$ and $SOCS_{200-300cm}$ in
$SOCS_{0-100cm}$, respectively.
The calculation of the SOC storage (Pg) for a region generally uses Eq. (8):
$$SOC_{storage} = \sum_{i=1}^{n} SOCS_i \times A \times 10^{-12} \qquad (8)$$
where $SOCS_i$ is the SOCS (kg·m$^{-2}$) at site $i$ and $A$ is the area (m$^2$) of each grid unit.
To test the predictive effects of the two machine learning methods, "leave-one-out"
cross-validation was conducted. We used the $R^2$ value, the mean error ($ME$, Eq. (9)),
and the root mean square error ($RMSE$, Eq. (10)) to evaluate the performance of the
prediction models.
$$ME = \frac{1}{n} \sum_{i=1}^{n} [D(x_i) - D^*(x_i)] \qquad (9)$$
$$RMSE = \sqrt{\frac{1}{n} \sum_{i=1}^{n} [D(x_i) - D^*(x_i)]^2} \qquad (10)$$
where $D(x_i)$ is the measured SOCS, $D^*(x_i)$ is the predicted SOCS, and $n$ is the number





of validation sites.
**3 Results**
**3.1 Performance of machine learning methods**
The results of the "leave-one-out" cross-validation showed that the RF model
exhibited a Pearson's correlation coefficient of 0.81, which was higher than that of the
GBRT model (0.79) and SVM model (0.77). In addition, the *RMSE* of the RF model
(3.01 kg·m$^{-2}$) was lower than that of the GBRT model (3.11 kg·m$^{-2}$) and SVM model
(3.21 kg·m$^{-2}$) for the upper 30 cm of the soil profile. These results suggest that the RF
model provides a better tool for predicting the spatial distribution of SOCS in the Third
Pole region. Moreover, in order to further discuss the simulation accuracy of the RF
model in this study, "leave-one-out" cross-validations were conducted for depth
intervals of 0–50 cm and 0–100 cm. The results revealed high $R^2$ as well as low *RMSE*
and *ME* values (Fig. 6).
**3.2 Storage and spatial distribution of soil organic carbon**
Figure 7 shows a large spatial variability of the SOCS across the Third Pole region,
whereby an overall decreasing trend can be observed from the southeast towards the
northwest. The wetland area in the eastern region of the Third Pole (Ruoergai) had the
highest predicted SOCS for a depth interval of 0–300 cm (> 32 kg·m$^{-2}$), whereas the
northern region (Qiangtang Plateau and Qaidam Basin) had the lowest SOCS (< 8 kg·m$^{-}$
$^2$). The estimated mean SOCS for the entire Third Pole region at depth intervals of 0–
30 cm, 0–50 cm, 1–100 cm, 0–200 cm, and 0–300 cm was 4.84 kg·m$^{-2}$, 6.45 kg·m$^{-2}$,
8.51 kg·m$^{-2}$, 11.57 kg·m$^{-2}$, and 14.17 kg·m$^{-2}$, respectively. Correspondingly, the total
estimated SOC storage was 15.79 Pg, 21.04 Pg, 27.75 Pg, 37.71 Pg, and 46.18 Pg at 0–
30 cm, 0–50 cm, 0–100 cm, 0–200 cm, and 0–300 cm, respectively (Table 2). In
addition, the SOCS decreased with increasing soil depth across the Third Pole region,
with 34.26% of the total SOC storage for a depth interval of 0–300 cm being contained
in the uppermost 30 cm, and only 17.89% in the 200–300 cm depth interval.
Compared with the area of seasonally frozen ground, the mean SOCS and total SOC
storage in the permafrost region were lower in each soil layer. The estimated amount of
SOC stored at a depth interval of 0–300 cm in the permafrost and seasonal frozen



ground zone were 21.69 Pg and 24.49 Pg, respectively, which accounted for 46.97%
and 53.03% of the total SOC pools, respectively.

The mean SOCS differed significantly among the various vegetation types ($P < 0.05$),

and were ranked as: forest > shrub > cropland > grassland > desert (Fig. 8). The
estimated SOC storage at a depth interval of 0–300 cm in forest, shrub, cropland,
grassland, and desert areas was 3.30 Pg, 0.85 Pg, 31.67 Pg, 9.77 Pg, and 0.59 Pg, thus
accounting for 7.15%, 1.84%, 68.58%, 21.57%, and 1.28% of the total, respectively.

According to the Harmonized World Soil Database, soils in the Third Pole region can

be divided into 21 main orders. Table 3 shows that the mean SOCS differed significantly
among various soil orders. The highest mean SOCS was for histosols (39.45 kg·m$^{-2}$),
which was ~3 times higher than that for leptosols (14.17 kg·m$^{-2}$), calcisols (11.50 kg·m$^{-}$
$^{2}$), cambisols (11.36 kg·m$^{-2}$), lithosols (12.91 kg·m$^{-2}$), and regosols (11.32 kg·m$^{-2}$). The
mean SOCS values of chernozems, greyzems, gleysols, podzoluvisols, and luvisols
were all > 20 kg·m$^{-2}$, whereas those of arenosols, salt flats, and solonchaks were all <
8 kg·m$^{-2}$. Due to the differences in the mean SOCS values and distribution area, the
total SOC storage of each soil order also differed significantly. The total SOC storage
of leptosols was ~25.41 Pg for a depth interval of 0–300 cm, thus accounting for 55.02%
of the total SOC pool in the area of frozen ground on the QTP, while other soil orders
were < 5 Pg.
**4 Discussion**

Due to the lack of systematic field observations, soil is still the part of the terrestrial

carbon cycle with the least amount of data, and the estimation of regional SOC pools
remains uncertain. Relatively few studies have estimated the SOC pool of the Third
Pole region. Most studies related to the Chinese part of the QTP (Tian et al., 2008; Wu
et al., 2008), or focused on the SOC storage of a certain vegetation type or certain area
(Wang et al., 2002). In addition, it is difficult to obtain data for deep soil horizons in the
Third Pole region due to complex terrain and harsh environment. Hence, most terrestrial
SOCS studies have focused on the shallow soil layer within 100 cm (Bai et al., 2010;
Fang et al., 1996; Yang et al., 2008), especially that of permafrost zones (Ding et al.,
2016; Mu et al., 2015; Wang et al., 2020; Zhao et al., 2018).



To date, few studies have, therefore, investigated the SOC storage and spatial patterns
in areas of seasonally frozen ground in the Third Pole region. Our study provides new
and more accurate data on SOC storage and spatial patterns for a depth interval of 0–3
m at a 1 $km^2$ scale over the Third Pole region, thus providing basic data for future
studies pertaining to Earth system modeling. We note that a lack of deep soil pits in
forest, shrub, and cropland ecosystems (Fig. S2) means some uncertainties in the
estimation of deep SOC pools remain; however, the collective area of these ecosystems
accounts for < 6% of the total area of the Third Pole region and may have a relatively
small influence on total SOC pools (Fig. A1). Regardless, there is a need for large-scale
soil surveys that include these areas in order to obtain more accurate information on the
SOC storage and distribution in the Third Pole region. Furthermore, regional SOC pools
are affected by many other factors, such as soil moisture (Wu et al., 2016) and grazing
activities (Zhou et al., 2017), which were not considered in our study due to lack of
high-resolution data with a high accuracy. Future work should consider the influence
of these factors on SOC at a regional scale to obtain more accurate datasets.
**5. Data availability**
The dataset of SOCS in the Third Pole region is available at the
https://doi.org/10.5281/zenodo.4293454 (Wang et al., 2020).
**6. Conclusions**
This study simulated the spatial pattern of the SOCS over the Third Pole region, and
systematically estimated the SOC storage (46.18 Pg) at a depth interval of 0–3 m for
the first time. Our results demonstrated that combining multi-environmental factors
with machine learning techniques (RF, SVM, and GBRT) can offer an effective and
powerful modeling approach for mapping the spatial patterns of SOC. Furthermore, this
study provided datasets of SOCS and SOC storage for permafrost and seasonally frozen
ground, as well as for various vegetation/soil types at different soil depths (0–30 cm,
0–50 cm, 0–100 cm, 0–200 cm, and 0–300 cm) across the Third Pole region. These
datasets can be used to modify existing Earth system models and improve prediction
accuracy, and also serve as a reference for policymakers to formulate more effective
carbon budget management strategies.





**Author contributions**

The study was completed with cooperation between all authors. Tonghua Wu and Xiaodong Wu conceived the idea of mapping the spatial distribution of the SOC across the Third Pole regions. Dong Wang conducted the data analyses and wrote the paper. All authors discussed the simulation results and helped revise the paper.

**Competing interests**

The authors declare that they have no conflict of interest.

**Acknowledgements**

This work was financially supported by the State Key Laboratory of Cryospheric Science (SKLCS-ZZ-2020), the National Natural Science Foundations of China (41690142, 41721091, 41771076, 41961144021, 41671070), and the CAS "Light of West China" Program.



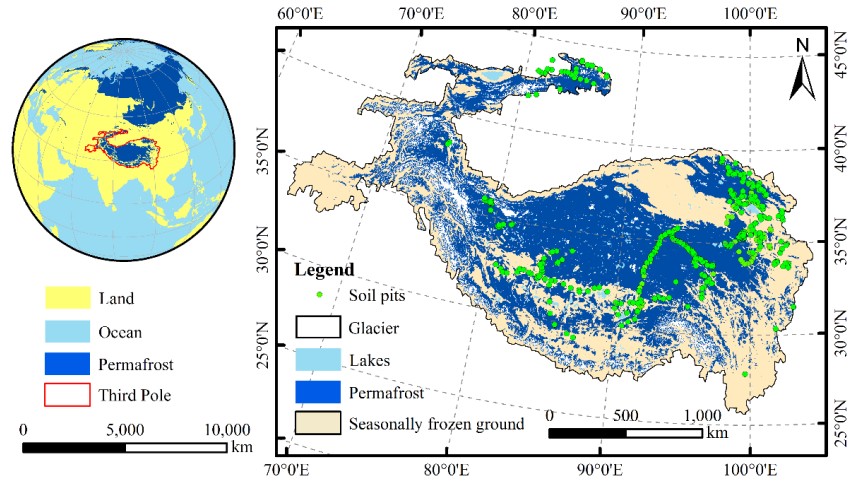

**Figure 1.** Distribution of soil pits in the Third Pole region (the frozen ground map is derived from
Obu et al., 2019).



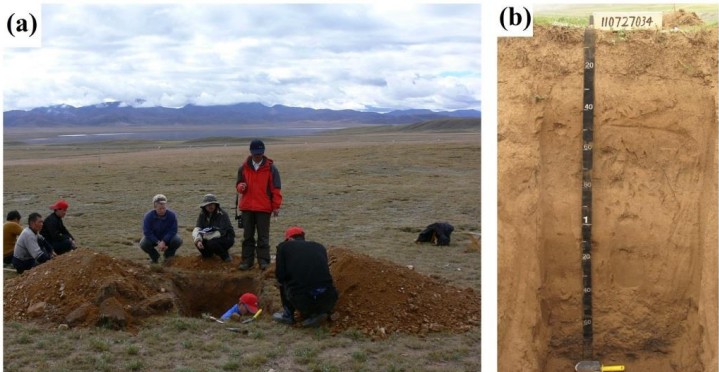


**Figure 2.** Field work photographs showing (a) soil sample collection, and (b) a soil profile.

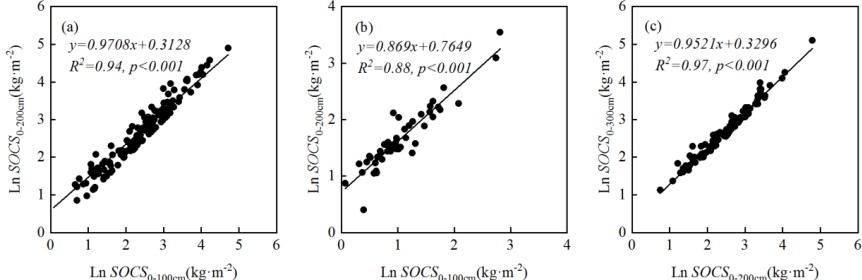


**Figure 3.** Extrapolation function of the SOCS between soil depth intervals of (a) 0–100 cm and 0–200 cm in grassland ecosystems, (b) 0–100 cm and 0–200 cm in desert ecosystems, and (c) 0–200 cm and 0–300 cm in grassland ecosystems.






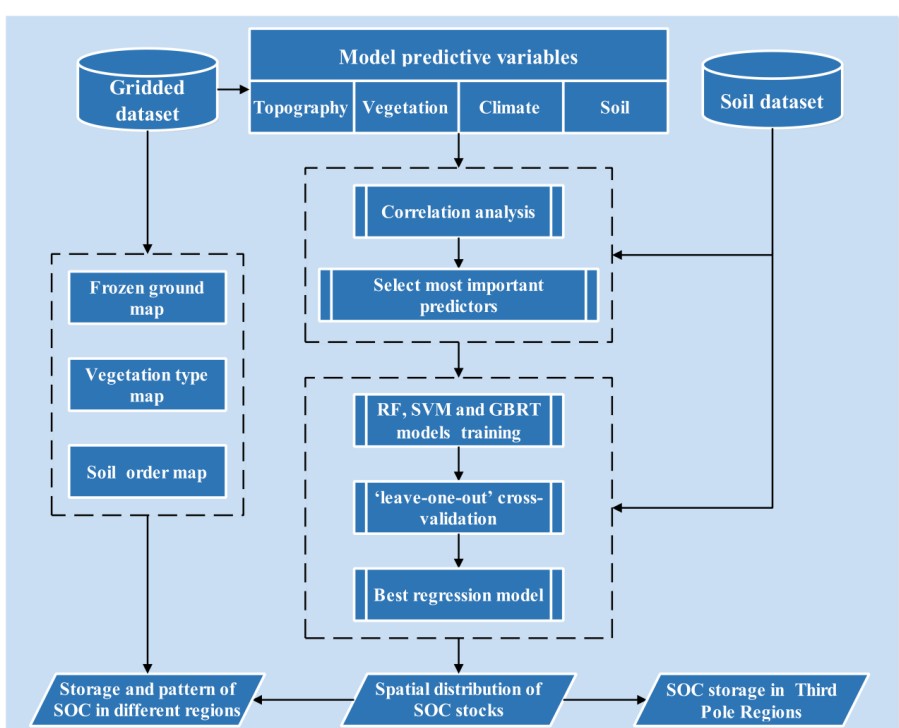

**Figure 4.** Workflow diagram for predicting SOCSs in this study. RF: random forest; SVM:

support vector machine; GBRT: gradient boosted regression tree.

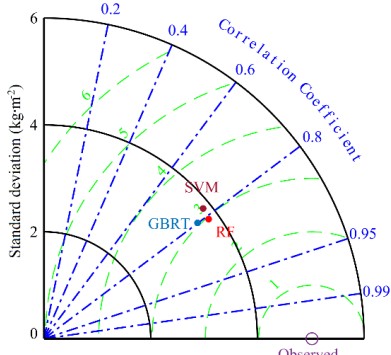

**Figure 5.** A Taylor diagram used to evaluate the model performance of random forest (RF), support vector machine (SVM), and gradient boosting regression tree (GBRT) models, which were used to predict the SOCS in the upper 30 cm of soil profiles across the Third Pole. The contour centered on the observed indicates the root-mean-square error (*RMSE*, kg·m⁻²) between the predicted value and observed value.





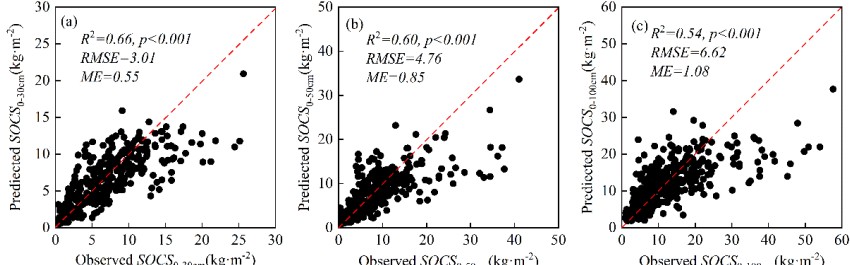

**Figure 6.** "Leave-one-out" cross-validation for the RF model used to predict the SOCS at (a) 0–30 cm, (b) 0–50 cm, and (c) 0–100 cm depth intervals.

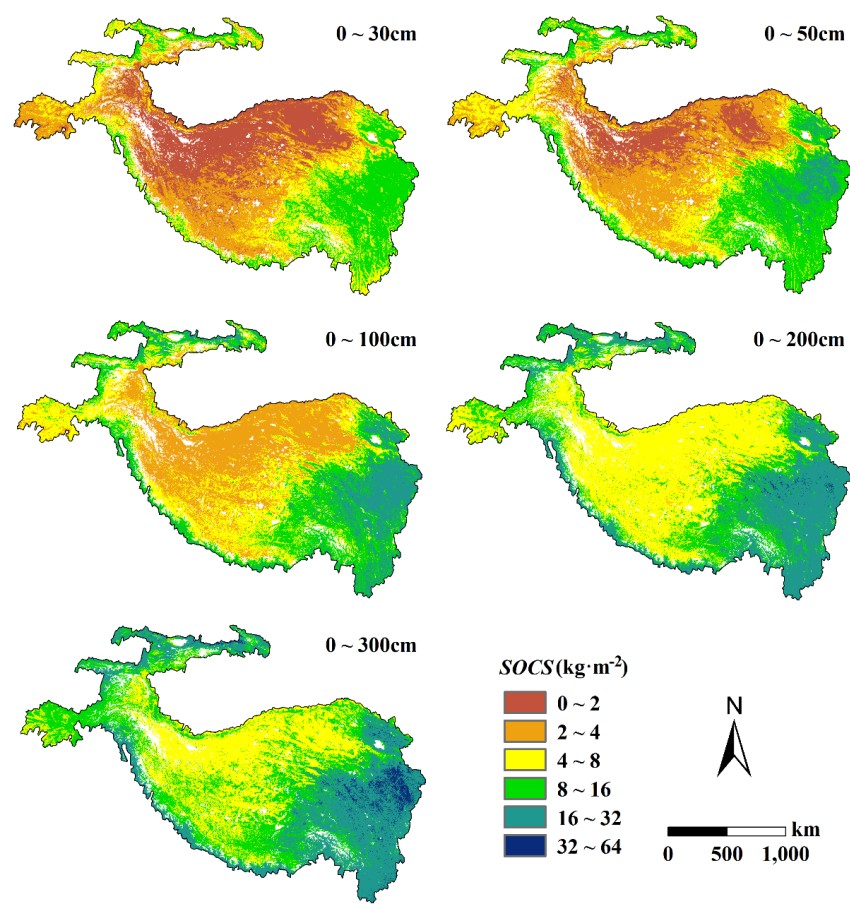


**Figure 7.** Spatial distribution of SOCS at different depth intervals over the Third Pole region.



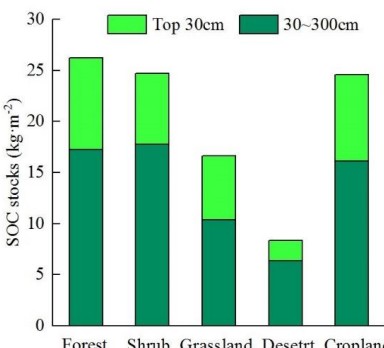


**Figure 8.** Summary of the estimated SOC stocks of different vegetation types in the Third Pole.




**Table 1** Summary of soil organic carbon datasets used in this study

| Number of samples | Depth interval | Period | Method | Source |
|---|---|---|---|---|
| 135 | 0–30 cm, 0–50, and 0–100 cm | 2001–2005 | Walkley-black method | Yang et al., 2010 |
| 30 | Genetic horizon | 2012–2013 | Walkley-black method | Song et al., 2016 |
| 93 | 0–100 cm | 2004–2014 | Walkley-black method | Xu et al., 2019 |
| 113 | 0–200 cm and 0–300 cm | 2013–2014 | Walkley-black method | Ding et al., 2016 |
| 200 | 0–200 cm | 2009–2013 | Walkley-black method | Field-measured |




**Table 2** Summary of the estimated mean SOC stocks and storages in permafrost and seasonally
frozen ground of the Third Pole

| Depth (cm) | SOC stock (kg·m$^{-2}$) | | | SOC storage (Pg) | | |
|---|---|---|---|---|---|---|
| | Permafrost | Seasonally frozen ground | Third Pole | Permafrost | Seasonally frozen ground | Third Pole |
| 0–30 | 4.13 | 5.56 | 4.84 | 7.61 | 8.63 | 15.79 |
| 0–50 | 5.72 | 7.16 | 6.45 | 10.53 | 11.12 | 21.04 |
| 0–100 | 7.28 | 9.70 | 8.51 | 13.41 | 15.06 | 27.75 |
| 0–200 | 10.25 | 12.88 | 11.57 | 18.88 | 19.99 | 37.71 |
| 0–300 | 12.52 | 15.40 | 14.17 | 21.69 | 24.49 | 46.18 |




**Table 3** Summary of the estimated mean SOC stock and storage of different soil orders in the
Third Pole

| Soil order | Area ($10^3$ km$^2$) | SOC stock (kg·m$^{-2}$) | | | SOC storage (Pg) | | |
|---|---|---|---|---|---|---|---|
| | | 0–30 cm | 0–100 cm | 0–300 cm | 0–30 cm | 0–100 cm | 0–300 cm |
| Leptosols | 1793.53 | 4.84 | 8.51 | 14.17 | 8.68 | 15.26 | 25.41 |
| Arenosols | 60.59 | 1.78 | 3.88 | 7.87 | 0.11 | 0.24 | 0.48 |
| Calcisols | 89.44 | 3.59 | 6.64 | 11.50 | 0.32 | 0.59 | 1.03 |
| Cambisols | 313.14 | 3.62 | 6.58 | 11.36 | 1.13 | 2.06 | 3.56 |
| Chernozems | 78.31 | 8.47 | 14.45 | 22.47 | 0.66 | 1.13 | 1.76 |
| Gypsisols | 61.64 | 1.36 | 3.36 | 7.40 | 0.08 | 0.21 | 0.46 |
| Greyzems | 16.26 | 9.61 | 15.44 | 23.82 | 0.16 | 0.25 | 0.39 |
| Gleysols | 71.98 | 11.71 | 18.73 | 29.04 | 0.84 | 1.35 | 2.09 |
| Kastanozems | 34.59 | 6.07 | 10.39 | 16.47 | 0.21 | 0.36 | 0.57 |
| Lithosols | 367.94 | 4.34 | 7.57 | 12.91 | 1.60 | 2.79 | 4.75 |
| Phaeozems | 44.01 | 4.77 | 8.45 | 13.68 | 0.21 | 0.37 | 0.60 |
| Luvisols | 156.35 | 9.37 | 15.71 | 25.04 | 1.46 | 2.46 | 3.92 |
| Solonchaks | 38.32 | 1.80 | 3.96 | 7.97 | 0.07 | 0.15 | 0.31 |
| Salt flats | 20.7 | 1.21 | 3.28 | 7.30 | 0.03 | 0.07 | 0.15 |
| Histosols | 3.62 | 13.33 | 27.36 | 39.45 | 0.05 | 0.10 | 0.14 |
| Anthrosols | 9.54 | 5.01 | 9.41 | 15.13 | 0.05 | 0.09 | 0.14 |
| Fluvisols | 8.97 | 3.06 | 5.78 | 10.19 | 0.03 | 0.05 | 0.09 |
| Regosols | 7.9 | 3.78 | 6.55 | 11.32 | 0.03 | 0.05 | 0.09 |
| Podzlos | 7.28 | 1.92 | 3.76 | 8.01 | 0.01 | 0.03 | 0.06 |
| Podzoluvisols | 2.96 | 8.90 | 13.57 | 21.60 | 0.03 | 0.04 | 0.06 |
| Rendzina | 1.94 | 5.26 | 9.48 | 16.14 | 0.01 | 0.02 | 0.03 |

*Soil orders with an area of < 1 km$^2$ were not included.



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
