# Peer review of "Soil organic carbon distribution for 0-3 m soil depth at 1-km resolution of the frozen ground in the Third Pole"

_Earth System Science Data, 2020_

## Author Comment (AC1)

**Reviewer 1#:**

The authors created high spatial resolution data of organic carbon distribution in the Third pole by compiling all the field data and using machine learning methods. The dataset can be very useful to help the scientific community to understand the carbon cycle. I found the paper is well organized. My major concern is that the authors should clearly explained what are the new findings in comparison with several previous reports on the plateau, e.g., Ding et al. 2016, Wang et al. 2020.

**Response:** Thanks very much for your review. We carefully read your comments and made substantial revisions according to your comments, we believe the quality of the manuscript has been greatly improved. Detailed responses are in blue, in-line with reviewer input below. In addition, we included Dr. Lin Zhao as a co-author because he made contributions to the intellectual merit of this work in our many previous discussions.

To clearly explain what are the new findings in comparison with several previous reports on the plateau, e.g., Ding et al. 2016, Wang et al. 2020. We have added these explanations in the methods section in the revised version as follows:

*In this study, we provided the new version of 1-km resolution maps of SOCS across the Third Pole at 0–300cm depth intervals, and largely makes up for the deficiencies of previous studies (Ding et al., 2016; Ding et al., 2019; Wang et al., 2020). On the one hand, our predictions have higher resolution than those studies. Take an example and focus on a $4.5 \times 10^4\ km^2$ local area situated in the Budongquan area of Qinghai province, China (Fig. 8). It can be seen from the excerpts of the map that our prediction is much more detailed than previous studies. Thus, our predictions better represented spatial variation of the SOCS across the Third pole region, especially for those regions with large heterogeneity. On the other hand, these reports most focused on the permafrost regions rather than the whole Third Pole (Ding et al., 2016; Wang et al., 2020). To date, few studies have investigated the SOC storage and spatial patterns in areas of seasonally frozen ground in the Third Pole region. In this study, we created high spatial*

*resolution data of SOCS distribution in the whole Third Pole by compiling all the field data and using machine learning methods, thus providing more accurate data than previous studies.*

[Figure]

**Figure 8.** Comparison of spatial details of the predictions with the previous studies: SOCS at 0–300 cm depth in the map excerpt of Budongquan area of Qinghai province, China. (a) Ding et al., 2016; (b) Ding et al., 2019; (c) Wang et al., 2020; (d) This study.

**Specific comments:**

**Figure 1 ,2 and 4 are never mentioned in the manuscript, please add reference mark in right place.**

Response: Thanks for your suggestion, we have added references mark in right place.

**Many SOC data were collected in this study, however, the data of China's national soil survey were not included in this manuscript, why?**

Response: During the collection process of SOC data, we referred to the information of the second China's national soil survey. however, this survey was conducted during 1978 - 1984, and the sampling points lacked accurate location information in Tibetan Plateau due to the limitation of technical means. In addition, most of them lacked data of coarse gravel content and bulk density, which could cause a large uncertainty source in SOC stocks calculation. In contrast, all soil points in this study were sampled after 2000, and all of them contained clear information on gravel content, bulk density and location, so the calculated SOC storage was more accurate.

**Please add the accuracy assessment in manuscript between your dataset and other global or regional SOC datasets, such as: SoilGrids and HWSD.**

Response: Thanks for your suggestion, we have added the accuracy assessment in the revised version between our dataset and other global SOC datasets (SoilGrids250m and WISE30sec) as follows:

[revised manuscript text omitted]

**L.33, L.216, and L.367. what is SOCSs?**

**Response:** Changed.

**L.302 & L.302. Change "in the area of" into "in the areas of"**

**Response:** Changed.

**L.223. Change "Fig. A1" to "Fig. S1" Also in L.323.**

**Response:** Changed.

**L.248. "To test the predictive effects of the two machine learning methods…", two or three?**

**Response:** Changed.

**L.289-291. "The estimated SOC storage at a depth interval of 0–300 cm in forest, shrub, cropland, grassland, and desert areas was 3.30 Pg, 0.85 Pg, 31.67 Pg, 9.77 Pg, and 0.59 Pg, thus accounting for 7.15%, 1.84%, 68.58%, 21.57%, and 1.28% of the total, respectively". Mismatching in different vegetation types and SOC storage values.**

**Response:** Changed.

**L.296. "lithosols" Inconsistent font.**

**Response:** Changed.

**L.302. Change "QTP" into "Third pole region"**

**Response:** Changed.

**L.313-314. Hence, most terrestrial SOCS studies have focused on the shallow soil layer within 100 cm …especially that of permafrost zones (Ding et al., 2016; Mu et al., 2015; Wang et al., 2020; Zhao et al., 2018). Wrong citations. The soil depth is deeper down to 100cm in those articles**

**Response:** We have deleted this sentence.

**L.475. Change "In" into "in".**

**Response:** Changed.

---

## Author Comment (AC2)

**Reviewer 2#:**

This is a very useful dataset that can be used and cited in the future. But the manuscript is written as a scientific research paper rather than a data description paper. I suggest the authors to restructure the paper to describe what the data actually contain. It is unclear to me whether the dataset contains the climate data, which are collected from some other source. So these things need to be more clearly described. Please also clarify what are modeled and what are in situ data. The title should also reflect this.

For the same reason, the introduction, results, and discussion sections are unnecessarily long and not very useful in actually understanding what the data are.

**Response:**

Thanks very much for your review. We carefully read your comments and restructure the paper according to your comments, we believe the quality of the manuscript has been greatly improved. Detailed responses are in blue, in-line with reviewer input below.

First, this dataset is the soil organic carbon distribution data at different soil depths (0–30 cm, 0–50 cm, 0–100 cm, 0–200 cm, and 0–300 cm) in the frozen ground area of the Third Pole region, and the dataset does not contain the climate data (precipitation and temperature), which are the environmental factors for constructing machine learning models in this study. Similarly, our dataset also does not contain the situ data, which are the input data for constructing machine learning models in this study. Therefore, the original title "Soil organic carbon distribution for 0-3 m soils at 1 km$^2$ scale of the frozen ground in the Third Pole Regions" reflects the information of the data subject (soil organic carbon), depth intervals (0-3 m), spatial resolution (1 km$^2$) and study area (Third Pole Regions).

Second, we have added a detailed description of the dataset in the **Data availability** section (section 5) to clarify the dataset information in this study. We have added these explanations in the methods section in the revised version as follows:

*The datasets of SOC stocks distribution in GeoTiff format are available at https://doi.org/10.5281/zenodo.4293454 (Wang et al., 2020). The file name is "TP-SOC-d.tif", where d represents soil depth, for example, "TP-SOC-30.tif" represents the spatial distribution of SOC stocks in the Third Pole regions of the upper 30 cm depth interval.*

Thirdly, we have truncated some content in the introduction, results and discussion sections, which are not very useful in actually understanding what the data are. In addition, we have added some subtitles and a more detailed description in the collecting and processing of soil sample data to improve the quality of the **Materials and Methods** section.

Finally, we have added the accuracy assessment in the revised version between our dataset and other global SOC datasets (SoilGrids250m and WISE30sec) in the discussion sections to further improve this article.

Minor comments:

**Please describe how and why the sampling area (soil pits) were selected.**

**Response:**

Because of harsh natural conditions and the inaccessibility of traffic in the Third pole regions, the soil samplings from earlier studies were conducted along the major roads, which unable to represent the environmental characters in large areas of the Third pole. Therefore, we conducted a large-scale field sampling from 2009 to 2013, covering all

major climatic regions and vegetation types across the plateau, including the large unpopulated area with harsh natural conditions.

The setting of sampling points for soil survey is widely representative, with at least one sampling point for each soil/geomorphic type. In areas with strong environmental heterogeneity, the layout of soil pits based on different parts of medium topography or micro-topography, especially in mountainous area with complex terrain. Moreover, the layout of sampling points avoids special sections such as roads, railways, engineering facilities, etc., which are greatly affected by human interference activities. Therefore, combined with the available published data and field investigated data, the 458 soil pits (depth of 0–1 m) and 114 soil cores (depth of 0–3 m) in this study can represent the vegetation types and characters in large areas of the Third pole.

Therefore, we have added the **Table 2** in **Data Processing** (section 2.2.1 in Manuscript) to display the number of soil sample points of different vegetation types in Third pole region.

**Table 2** Number of soil sample points of different vegetation types in Third pole region

| Vegetation types | Forest | Shrub | Grassland | Desert | Cropland |
|---|---|---|---|---|---|
| Number | 10 | 22 | 371 | 49 | 6 |

**- How did the authors deal with gravel and what was the gravel %?**

**Response:**

The weak chemical and biological processes on the third pole resulted in the widespread existence of gravel in soil. Missing the gravel data will affect the estimation of bulk density, which leads to the erroneous estimation of SOC storage. In our study, all soil samples for carbon analysis were air-dried, handpicked to remove plant detritus, and then sieved through a 2mm mesh to calculate the volume percentage of the gravel. In addition, the rest of SOC data, which obtained from Yang et al. (2010), Song et al.

(2016), Xu et al. (2019) and Ding et al. (2016), was treated with the similar method as our experimental procedure.

According to the statistics of 200 soil samples obtained from the field (Table 5), it was found that the gravel content of grassland soil increased with the depth of 2-m profiles, from 4.65% in 0-30cm depth interval to 28.17% in 100-200cm depth interval. The volume percentage of soil gravel in desert areas is overall higher than that in grassland areas, especially in the top 50cm, but no significant change with soil depth increased in the 2-m profiles. It should be noted that the soil sample points in forest, scrub and cropland areas are derived from the literatures, and the volume percentage of gravel is unclear.

**Table 5** Statistical description of the gravel content at different depths based on the 200 soil profiles.

| Vegetation types | Gravel content (%) | | | |
|---|---|---|---|---|
| | 0–30 cm | 30–50 cm | 50–100 cm | 100–200 cm |
| Grassland | 4.65 ± 1.98 | 11.91 ± 3.59 | 24.89 ± 4.04 | 28.17 ± 4.53 |
| Desert | 18.56 ± 4.15 | 21.6 ± 4.76 | 20.84 ± 4.36 | 21.07 ± 4.69 |

---

## Author Response (AR2)

**Reviewer 1#:**

The reviewers all agree that it would be crucial to have a dataset representing the soil organic carbon for the Tibetan Plateau, where a large amount of organic carbon is stored and could be a risk driving by climate change. The dataset provides a high-resolution and consistent dataset for the SOC in the region and could be helpful to many applications. I believe the authors did a good job presenting the dataset in the manuscript and have properly addressed the concerns and suggestions from the review in the revision. However, I did notice issues in the current manuscript and suggest the authors could address those to further improve the manuscript.

Thank you very much for your time and effort regarding our manuscript. We have carefully revised the manuscript according to your comments. Detailed responses are in blue, in-line with reviewer input below.

**Specific comments:**

1) The title can be further optimized. The term "scale" usually refers to distance instead of area, for example, 1 km stead of 1 km2. I would suggest the authors add "depth" to "0-3 m" to clarify what it refers to. The word "Regions" seems redundant following "the Third Pole".

**Response:** Thanks for your suggestion, we have changed the title to "Soil organic carbon distribution for 0-3 m soil depth at 1-km resolution of the frozen ground in the Third Pole".

**2) There are many writing and formatting issues, for example, "the Third Pole" has been written as "the third pole", "the Third pole"; missing hyphen in "1-km resolution" and "8-day", ... I would suggest the authors carefully check for grammar and formatting issues through the manuscript. I think it would be very helpful if the authors could consider using help from professional language services to improve the writing and address these issues.**

**Response:** Thanks for your suggestion, we have carefully checked and modified the format errors in the manuscript. In addition, this manuscript has been professionally polished by a language editing service, we believe the quality of manuscript has improved, and the proof from the editing service as follows:

[Figure]

**Line 113. The phase is not accurate because the desert is usually not considered as a type of vegetation.**

**Response:** Thanks for your suggestion, we have changed the "vegetation" to the "ecosystems".

**Line 192. "from 2010" change to "for 2010".**

**Response:** Changed.

\#3) The notations SOCS 0-100cm/0-200 cm in equations 2 and 3 are the same and could be confusing. I would suggest adding proper subscripts to distinguish the notations for grassland and desert ecosystems. The same suggestion also applies to Figure 4.

**Response:** Thanks for your suggestion, we have added proper subscripts to distinguish the notations for grassland and desert ecosystems in equations 2, 3 and 4.

$$\ln SOCS_{G(0-200cm)} = 0.9708 \times \ln SOCS_{G(0-100cm)} + 0.3128 \tag{2}$$

$$\ln SOCS_{D(0-200cm)} = 0.8690 \times \ln SOCS_{D(0-100cm)} + 0.7649 \tag{3}$$

$$\ln SOCS_{G(0-300cm)} = 0.9521 \times \ln SOCS_{G(0-200cm)} + 0.3296 \tag{4}$$

where $\ln SOCS_{G(0-100cm)}$, $\ln SOCS_{G(0-200cm)}$ and $\ln SOCS_{G(0-300cm)}$ are the natural logarithms of the SOC stocks ($kg \cdot m^{-2}$) in grassland ecosystems at the depth intervals of 0–100 cm, 0–200 cm, and 0–300 cm, respectively; likewise, $\ln SOCS_{D(0-100cm)}$ and $\ln SOCS_{D(0-200cm)}$ are the natural logarithms of the SOC stocks ($kg \cdot m^{-2}$) in desert ecosystems at the depth intervals of 0–100 cm and 0–200 cm, respectively.

\#The colors in Figure 3 and Figure 5 can be improved. The color in Figure 1 would be proper for a dark background presentation but making it hard to read on a white background paper. For Figure 5, I would suggest using a stronger color to replace the light green, which is hard to read.

**Response:** Thanks for your suggestion, we have modified the Figure 1, Figure 3 and Figure 5, as shown below.

[Figure]

**Figure 1.** Distribution of soil pits in the Third Pole region (the frozen ground map is derived from

Obu et al., 2019).

[Figure]

**Figure 3.** Workflow diagram for predicting SOCSs in this study. RF: random forest; SVM: support vector machine; GBRT: gradient boosted regression tree.

[Figure]

**Figure 5.** A Taylor diagram used to evaluate the model performance of random forest (RF), support vector machine (SVM), and gradient boosting regression tree (GBRT) models, which were used to predict the SOCS in the upper 30 cm of soil profiles across the Third Pole. The contour centered on the observed indicates the root-mean-square error ($RMSE$, kg·m$^{-2}$) between the predicted value and observed value.

---

## Author Response (AR3)

**Reviewer 1#:**

Comments to the Author:

I am sorry for asking for another minor revision. I appreciate the effort on improving the manuscript. It's much better now, but I still wonder if you could tweak the title for one more time? The current one is ill-structured because "1-km resolution" is supposed to describe the distribution instead of the soil depth, and it could be confusing when put it between the "soil depth" and "of the frozen ground".

My suggestions are:

Soil organic carbon distribution at 1-km resolution for 0-3 m depth of frozen ground in the Third Pole

Or

A 1-km resolution soil organic carbon dataset for 0-3 m depth of the Third Pole frozen ground

Or simplify it to:

A 1-km resolution soil organic carbon dataset for frozen ground in the Third Pole

Please do not feel obligated to use any of the above suggestions, but I do hope that you could give it one more thought, making it correct and excellent in every way before publication.

**Response:** Thanks for your suggestion, we have changed the title to "A 1-km resolution soil organic carbon dataset for frozen ground in the Third Pole".